# Fitness Trainers’ Educational Qualification and Experience and Its Association with Their Trainees’ Musculoskeletal Pain: A Cross-Sectional Study

**DOI:** 10.3390/sports10090129

**Published:** 2022-08-29

**Authors:** Sohel Ahmed, Mamunur Rashid, Abu-sufian Sarkar, Mohammad Jahirul Islam, Rahemun Akter, Masudur Rahman, Shahana Islam, Devjanee Sheel, Sarwar Alam Polash, Mahfuza Akter, Shayed Afride, Manzur Kader

**Affiliations:** 1Department of Physiotherapy & Rehabilitation, Mount Adora Hospital, Akhalia, Sylhet 3100, Bangladesh; 2Physiotherapy Research Foundation, Akhalia, Sylhet 3100, Bangladesh; 3Department of Public Health and Sports Science, Faculty of Occupational and Health Sciences, University of Gävle, Kungsbacksvägen, 80176 Gävle, Sweden; 4Bashundhara Kings Football Club, Block: D, Bashundhara R/A, Dhaka 1229, Bangladesh; 5Department of Physical Medicine and Rehabilitation, M A G Osmani Medical College Hospital, Sylhet 3100, Bangladesh; 6Dream Physiotherapy and Rehabilitation Center, Paikpara, Barhmanbaria 3400, Bangladesh; 7Department of Physiotherapy, CB Hospital Ltd., Polashpole, Satkhira 9400, Bangladesh; 8Department of Physiotherapy, Shiekh Fazilatunnessa Mujib Memorial KPJ Specialized Hospital, Gazipur 1700, Bangladesh; 9Gonoshasthay Somajvittik Physiotherapy College, Mirzanagar, Saver, Dhaka 1344, Bangladesh; 10Department of Physiotherapy, Caritas Bangladesh, Ashulia, Saver, Dhaka 1344, Bangladesh; 11Department of Medicine, Solna, Clinical Epidemiology Division, Karolinska Institutet, 17176 Stockholm, Sweden

**Keywords:** exercise, fitness trainer, fitness center, injuries, pain

## Abstract

This is a cross-sectional study that examined the association between fitness trainers’ educational qualifications and experience, and the risk of their trainees’ musculoskeletal pain. The study included 1177 trainees (aged 15–60 years) from 74 fitness centers in Bangladesh. Data were collected by using the Nordic musculoskeletal questionnaire, including potential confounders such as demographic factors (e.g., age, occupation), and training-related factors (e.g., workout knowledge, overweight lifting). Multiple logistic regression was performed for a binary outcome (pain—yes or no), and a generalized linear model was fitted for the ordinal outcome (pain—sites of the body). The trainers’ lower experience (no or ≤1 year) was associated with higher odds of their trainees’ musculoskeletal pain (OR: 2.53, 95% CI: 1.18–5.44) compared to trainers with >5 years of experience; however, no association was found between the trainers’ education and the risk of their trainees’ musculoskeletal pain, after controlling for potential confounders. Similarly, the trainees trained by trainers with lower experience had more than two-time the risk of having pain in different sites (IRR: 2.04, 95% CI: 1.50–2.79). The trainers’ experience may play a pivotal role in the trainees’ musculoskeletal pain. Further study is warranted in this regard.

## 1. Introduction

Physical activity helps to maintain good health and improve physical functioning and the quality of life [1]. Fitness centers are an important domain for physical activity, where the members like to exercise more often than they usually do at home [2]. A study of the level of physical activity among adults in 28 European countries reported that 59.1% of the respondents were highly physically active, which was achieved by vigorous exercise [3]. The Physical Activity Council’s overview report on the U.S. population reported that 67% of the population over the age of six participated in fitness sports in 2020 [4]. The Eurobarometer on sports and physical activity (2014) reported that 15% of the European population attended fitness centers in 2013 [5].

Fitness professionals help their trainees with various domains of exercise and training, using an individualized approach and assessment skills. They are expected to design safe and effective exercise prescriptions, including resistance training and cardiovascular endurance training programs [6]. Talpey and Siesmaa [7] argue that fitness professionals have a duty to the participants they serve to take reasonable steps to prevent injury and to act prudently when an injury occurs. If an injury is not treated initially, it can turn into a chronic musculoskeletal pain condition [8]. Musculoskeletal pain refers to acute or chronic pain which is associated with musculoskeletal disorders and affects the muscles, tendons, ligaments, and bones of the body [9].

A study based on focus group methodology reports that fitness professionals have a wide variety of backgrounds, some having a collegiate-type degree and/or a group fitness trainer certification course, or some having experience only [10]. A survey of 115 health fitness professionals suggests that a personal trainer should have up-to-date knowledge in anatomy, biomechanics, kinesiology, and basic injury management, with advanced knowledge in nutrition, testing protocol, health screening, and exercise prescription [11]. Furthermore, another study shows that personal trainer fitness-related knowledge improves with a bachelor’s degree and more rigorous certification, and the trainer’s educational degree is highly associated with the performance of their trainee in Olympic weight lifting [6]. We hypothesized that the instructors’ experience can help to deliver their clients safe and effective exercise programs and gym usage. This includes knowing how to instruct the clients on the exercises and to use the gym equipment. Using the instructors’ experience, they can ensure that the exercises go smoothly, without any injury or complete exhaustion. Psychologically, trainees react and push themselves in different ways when they know that an experienced and knowledgeable instructor is watching them while exercising. Therefore, there is a need to investigate the association between trainer experience and musculoskeletal pain.

Studies related to the injuries occurring during fitness activities exist in small numbers, and most of the studies focus only on the types of injury, such as injuries due to overexertion, crushing, falls, awkward landing [12], the epidemiology of an aerobic dancing injury [13], the epidemiology of a weight training injury [14], or the frequency [15,16] in which they occur. Few studies have given attention to the specific causes of those injuries in the fitness center context [12,17]. Fitness professionals play a vital role in injury prevention for their trainees [7]. Yet, to the best of our knowledge, no study has investigated the association between the trainers’ educational qualifications and experience, and the trainees’ musculoskeletal pain. Therefore, this study aims to find out the association between fitness professionals’ educational qualifications and experience, and the musculoskeletal pain among their trainees.

## 2. Materials and Methods

### 2.1. Study Design and Population

A cross-sectional survey was conducted, based on data collected between 25 October and 15 December 2021, among the fitness trainers and their trainees. 

### 2.2. Participant’s Recruitment Criteria

Fitness trainees, both males and females, who were regular clients and had completed at least one month of fitness center membership, aged from 15 to 60 years, and who were willing to participate, were included in this study. The exclusion criteria were: professional athletes (who have a high chance of sports-related injuries that may hinder the study results), participants with a history of traumatic injuries within the past six months, age-related musculoskeletal disorders, and any other neurological disorders. The selection criteria for the trainer were: the lead trainer of the fitness center, whose work experience was a minimum of one month at the center, and who was willing to participate in this study.

### 2.3. Sample and Sample Size

Approximately 1300 trainees were interviewed. We received 1211 trainees’ data, and 33 participants were excluded as they did not answer all the questions; thus, a total of 1177 trainees were included in the final analysis. A flowchart diagram of the number of fitness centers and the numbers of trainees in five metropolitan cities are presented in Figure 1. 

The required sample size was calculated, based on the primary outcomes of the fitness trainees. A margin of 5% error, 95% confidence level, and 50% response distribution were used to calculate the required sample by using the formula: n = Zα^2^ P (1 − P)/d^2^. Using a 10% error rate, the study required a minimum of 1068 samples for this study. 

### 2.4. Survey Administration

An interviewer-administered questionnaire was carried out for this survey. A total of eight data collectors who graduated in physiotherapy were appointed for data collection. The medium of the interview was in the Bangla language. First, the data collector asked the question to the respondent and, when the answer was given, the data collector repeated the answer to the respondent for confirmation. The interview was carried out in two separate parts: first, the data collector collected the data from the lead trainer about their experience and educational background; after that, interviews were carried out among their trainees. 

### 2.5. Variables Measurement

#### 2.5.1. Explanatory Variables

This section included a questionnaire related to the trainer’s educational background (whether they had any fitness-related degree or certification apart from their professional fitness training courses) and working experience (total working experience as a fitness trainer in a fitness center).

#### 2.5.2. Outcomes 

The current (last 7 days) history of musculoskeletal pain was assessed by asking the question: “do you currently have any pain in your body?”. If the participant responds yes; the site of pain was identified by using the Nordic musculoskeletal disorder questionnaire. The questionnaire contains eight various sites of the body where the pain exists, including the neck, shoulder, elbow, wrist/hand, back, hip/thigh, knee, and ankle/foot. The validity of the Nordic musculoskeletal questionnaire has already been established in previous literature [18]. Kuorinka. I. et al. tested the validity against the clinical history of the Nordic musculoskeletal questionnaire and found a range of 0–20% disagreement [19]. A total of 7 days of clinical examination found sensitivity ranging between 66% and 92%, and specificity between 71% and 88%, for comparing pain in the body [20].

#### 2.5.3. Covariates

The following information included in the questionnaire were used as covariates in this study: demographic information (such as age, gender, BMI, and occupation), training-related information (such as workout knowledge, overweight lifting, and wrong holding). 

### 2.6. Survey Validation

The initial draft of the survey was validated by a panel of experts (one professor, two associate professors, two consultants in sports physiotherapy, and two epidemiologists). The draft survey was revised and adjusted following suggestions from the experts. The revised questionnaire was tested on a small, targeted sample population to check whether it was misleading or not. While testing the questionnaire, the response time was monitored. It took a maximum of 15 min to complete the survey. A two-day training program was arranged for the data collectors. Before the main study, a pilot study was conducted (each data collector collected 10 data samples, which are not included in the main analysis) to observe the capacity of the data collector and to check whether the questionnaire was misleading or not. Any necessary corrections were made after the pilot study. The final draft of the questionnaire contains four sections (Appendix A), and a total of 15 questions in Appendix A.

### 2.7. Ethical Statement

The study protocol was approved by the ethical review board of Mount Adora Hospital, Sylhet, Bangladesh (Approval No: MAH/ ERB: 21/013). We strictly followed the ethical guideline delivered by the Declaration of Helsinki (Revised 2013). The nature and purpose of this study were described in brief to the participants; after that, written informed consent was obtained (in case of minors, written assent was obtained from the participants, and informed consent was obtained from their guardians).

### 2.8. Statistical Analysis

Before the data analysis, the data were cleaned, managed, and checked for internal consistency. The descriptive statistics were expressed as the frequency, percentage, mean, and standard deviation. Pearson’s Chi-square test or Fisher’s exact test were used to see whether there was a difference between no pain and the presence of pain, with regard to all the selected variables. A multiple logistic regression model (LRM) was performed to estimate the association of the trainers’ degrees and experience with pain (yes = presence of pain or no = no pain), using the odds ratios (OR) and a 95% confidence interval (CI). The variables that significantly differed between no pain and the presence of pain in the bivariate analysis were controlled for in the adjusted analysis. A generalized linear model (GLM) was computed to measure the association between the trainers’ degrees and experience, and the sites of pain, using the relative risk (RR) and a 95% confidence interval (CI). Nagelkerke’s R^2^ was used to measure the overall predictive ability of the selected factors of pain in the LRM and GLM. The level of significance was set at *p* < 0.05 in all tests. All data analyses were performed using version 27 of the IBM SPSS statistical software package.

## 3. Results

### 3.1. Demographic Details of the Trainees

A total of 1211 trainees were interviewed from 74 different fitness centers across five selected areas of Bangladesh. After the final checking, 1177 trainees were included (34 incomplete forms were excluded) for the final analysis. The mean age, weight, and height of the trainees were 26.7 ± 6.8 years, 69.4 ± 10.8 kg, and 168.5 ± 7.3 cm, respectively. Most of the trainees were males (92.3%). Among the participants, 44% of the participants reported the current prevalence of musculoskeletal pain. The prevalence of musculoskeletal pain reported in different areas of the body were: the neck 42 (3.6%), shoulder 131 (11.1%), elbow 46 (3.9%), wrist/hand 65 (5.5%), back 202 (17.2%), hip/thigh 74 (6.3%), knee 85 (7.2%), and ankle/foot 53 (4.5%). A total of 24.7% of the participants performed over-exercise, followed by overweight lifting (22.1%), the wrong holding of the equipment (19.6%), and lack of workout knowledge (15.2%); almost half of the trainees were of a healthy weight (48.34%), and more than 50% of the trainees performed both aerobic and non-aerobic exercise. The demographic details of the trainees are presented in Table 1.

### 3.2. Association of Trainers’ Educational Qualifications and Experience with Trainees’ Musculoskeletal Pain 

Table 2 shows the association of the trainers’ educational qualification and experience with the trainees’ musculoskeletal pain. In the unadjusted analysis, the trainees who were trained by the trainers with no educational degree had higher odds of having musculoskeletal pain compared to trainees who were trained by the trainers with a bachelor’s/master’s degree; however, the results did not remain significant in the adjusted analysis. Similarly, the trainees who were trained by trainers who had work experience of no or <1 and 1–5 years were more likely to be experienced with musculoskeletal pain, in comparison to the trainees trained by the trainers with more than 5 years’ experience. However, only the result of the trainees who were trained by trainers who had no or <1 year of work experience remained significant (OR: 2.53, 95% CI: 1.18–5.44; *p* = 0.02).

Table 3 indicates the incidence rate ratio (IRR) of the trainers’ degrees and experience relative to the trainees’ pain in different sites of the body. In terms of the trainers’ degrees, the results indicate that trainees who had trainers with certificates and diploma degrees were increased with more incidence rate, i.e., 32% and 33% respectively, for site pain, compared to the trainees who had trainers with bachelor’s/master’s degrees, in the unadjusted analysis. However, the results did not remain significant after controlling for age, gender, BMI, occupation, type of exercise, and training-related factors, such as workout knowledge, overweight lifting, and wrong holding. The trainees trained by trainers who had experience of no or < 1 year and 1–5 years’ experience had 1.95- and 1.21-times, respectively, the rate of having pain in different sites compared to the trainees who had trainers with >5 years’ experience, in the unadjusted analysis. The results did remain significant for both categories—no or <1 year (IRR: 2.04, 95% CI: 1.50–2.79) and 1–5 years’ experience (IRR: 1.26, 95% CI: 1.05–1.51) in the adjusted analysis.

## 4. Discussion

This cross-sectional study aimed to examine the association between fitness trainers’ educational qualifications and experience and the risk of musculoskeletal pain among their trainees. The study found a significant association between the experience of the trainers and the musculoskeletal pain among their trainees, after controlling for potential confounders. However, the trainers’ educational qualifications did not play any significant role in this population.

The present study suggests that the trainees who worked out under the experienced trainers experienced less pain than the trainees with the less experienced trainers. This finding is in line with previous research, demonstrating that personal trainers, sports coaches, and strength and conditioning coaches’ knowledge of injury management plays a vital role in the prevention of injuries [7,11,21,22], which otherwise could, in turn, lead to developing musculoskeletal pain. Another study observed a significant association between the prevalence of injury and a lack of supervision [23]. An explanation for this result could be that experienced and qualified trainers may have basic knowledge about injury prevention, injury management, and exercise prescription, which could help in the safety of their trainees while they are working out [11] Personal trainers, sports coaches, and strength and conditioning coaches’ knowledge of injury management plays a vital role in injury prevention [7,11,21,22]. A significant association was observed between the prevalence of an injury and a lack of supervision, and the prevalence of injury and training multiple times in a day [23].

The role of strength and conditioning coaches to prevent sports injuries is well known in the literature, as they observe the exercise technique and modify it accordingly [7]. Sports coaching knowledge helps to translate into the prevention of injury among adolescents who are practicing netball and rugby, according to the literature. Coaches play a vital role in musculoskeletal injury prevention, as they have advanced knowledge regarding sports-specific injuries and fitness/conditioning, and they develop individual techniques to enhance the talent of their players [21]. Inappropriate training programs and a rapid raise in training intensity are more likely to cause musculoskeletal injuries [24]. Personal trainers’ knowledge of injury management, injury prevention, exercise prescription, and exercise training is highly associated with their educational degree [11]. However, in our study, most of the trainers who had >5 years of experience were not highly educated. One of the possible reasons for this is that, in Bangladesh, only three universities (Jessore University of Science and Technology, Uttara University and National University of Bangladesh) provide graduation-level degrees in physical education, and Bangladesh Krira Shikkha Protishtan provides a post-graduation diploma in sports training. There is no other institution that directly provides professional fitness trainer-training courses in Bangladesh. Fitness professionals in Bangladesh are facing standard educational degrees, which might be the reason that education was not significantly associated with these results. The more experienced and qualified trainers would naturally have attained more knowledge in injury prevention and exercise prescription and, therefore, can implement safer techniques when training their trainees, which helps to reduce the chances of injury and create a safer environment in their fitness center.

## 5. Strengths and Limitations

One of the strengths of the present study is that it used face-to-face interviewer-administered questionnaires, which helped to reduce non-responses and any misclassification bias, and thereby increases the precision and validity of the data. The inclusion of the nationwide population may help to increase the generalizability of the findings. Nevertheless, this study also has several limitations. This study was a non-random convenience sample from fitness centers in five metropolitan cities in Bangladesh. As such, the results of the trainers’ details in this study do not represent all the trainers in the country. This study included only the lead trainer of the fitness center; other trainers in the center with less working experience who are also providing instruction to the trainees might have a negative effect on the incidence of musculoskeletal pain and injuries. Since our research used self-reported data, we cannot eliminate the response bias in this study. In this study, we only interviewed the trainers about their experience and academic degree. Other variables, such as knowledge about basic injury management, safe and effective exercise prescription, and the safety rules of exercising equipment, were not assessed, which might influence the study. 

## 6. Conclusions

The trainers’ experience is associated with the risk of the trainees’ musculoskeletal pain. However, this study found no significant results in terms of the trainers’ education and the trainees’ musculoskeletal pain. Thus, care should be given by ensuring standard experience when appointing fitness trainers to the center. Yet, further research is needed in this regard, based on new samples that would be equally disseminated in terms of the trainers’ educational qualification and experience. 

## Figures and Tables

**Figure 1 sports-10-00129-f001:**
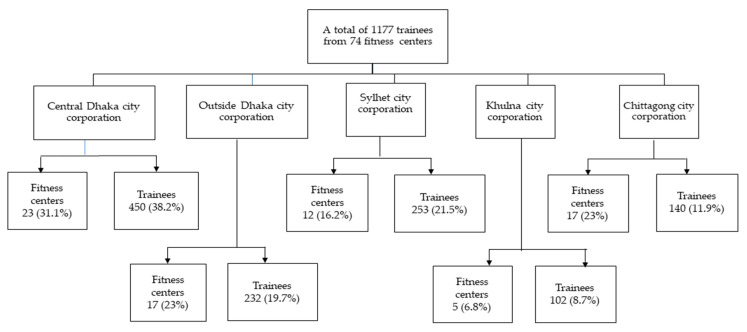
A flowchart diagram of the study participants.

**Table 1 sports-10-00129-t001:** Trainees’ Demographic Characteristics.

Variable		n (%)	Missing Value (n)	No Pain, n (%)	Presence of Pain, n (%)	*p*-Value
Total sample		1177 (100)		659 (56)	518 (44)	
Gender	Male	1086 (92.3)	0	595 (54.8)	491 (45.2)	0.004
	Female	91 (7.7)		64 (70.3)	27 (29.7)	
Age group	15–25	586 (49.8)	0	380 (64.8)	206 (35.2)	<0.001
	26–35	493 (41.9)		246 (49.9)	247 (50.1)	
	>35	98 (8.3)		33 (33.7)	65 (66.3)	
BMI category ^A^	Underweight	317 (26.93)	241	230 (72.6)	87 (27.4)	<0.001
	Healthy weight	569 (48.34)		285 (50.1)	284 (49.9)	
	Overweight/Obese	50 (4.24)		19 (38)	31 (62)	
Occupation	Student	473 (40.2)	0	318 (67.2)	155 (32.8)	<0.001
	Employed	634 (53.8)		311 (49.1)	323 (50.9)	
	Homemaker/Unemployed	70 (6)		30 (42.9)	40 (57.1)	
Purpose of GYM joining	Losing weight	268 (22.7)	0	101 (37.7)	167 (62.3)	<0.001
	Physical fitness	626 (53.2)		373 (59.6)	253 (40.4)	
	Bodybuilding	269 (22.9)		181 (67.3)	88 (32.7)	
	Recreation	14 (1.2)		4 (28.6)	10 (71.4)	
Hours spent working out in a day	<1 h	62 (5.3)	0	28 (45.2)	34 (54.8)	0.189
	1 h	574 (48.8)		321 (55.9)	253 (44.1)	
	2 h or more	541 (45.9)		310 (57.3)	231 (42.7)	
Type of exercise ^B^	Aerobic	187 (15.9)	0	58 (31.1)	129 (68.9)	<0.001
	Non-aerobic	389 (33.0)		230 (59.1)	159 (40.9)	
	Both	601 (51.1)		371 (61.7)	230 (38.3)	
Over exercise	No	886 (75.2)	0	536 (60.5)	350 (39.5)	<0.001
	Yes	291 (24.8)		123 (42.3)	168 (57.7)	
Wrong holding ^C^	No	947 (80.4)	0	570 (60.2)	377 (39.8)	<0.001
	Yes	230 (19.6)		89 (38.7)	141 (61.3)	
Overweight lifting	No	918 (77.9)	0	569 (62)	349 (38)	<0.001
	Yes	259 (22.1)		90 (34.7)	169 (65.3)	
Lack of workout knowledge	No	997 (84.8)	1	624 (62.6)	373 (37.4)	<0.001
	Yes	179 (15.2)		34 (19)	145 (81)	
Education of the trainer ^D^	No educational degree	450 (38.3)	0	274 (60.9)	176 (39.1)	0.004
	Certificate course	417 (35.4)		219 (52.5)	198 (47.5)	
	Diploma degree	175 (14.8)		83 (47.4)	92 (52.6)	
	Bachelor/Master degree	135 (11.5)		83 (61.5)	52 (38.5)	
Experience of the trainer ^E^	No or <1 year	61 (5.2)	0	19 (31.1)	42 (68.9)	<0.001
	1–5 years	609 (51.7)		330 (54.2)	279 (45.8)	
	>5 years	507 (43.1)		310 (61.1)	197 (38.9)	

^A^ BMI category: <18.5 = underweight, 18.5–24.9 = healthy weight, ≥25 = overweight/obese. ^B^ Type of exercise: aerobic: running, cycling, etc.; non-aerobic: sprinting, weight lifting, etc. ^C^ Wrong holding: the faulty technique of holding equipment while exercising. ^D^ Certificate course: having a fitness-related authorized certificate/diploma; 1–3 years’ degree related to fitness training, bachelor’s/master’s: 4-year graduation degree/1-year master’s degree, ^E^ Working in a fitness center as a trainer.

**Table 2 sports-10-00129-t002:** Multiple Logistic Regression Model: Risk Factors of the Pain of the Trainee.

Variables	Unadjusted Analysis		Adjusted Analysis ^1^	
	OR (95% CI)	*p*-Value	OR (95% CI)	*p*-Value
Education of the trainer ^A^				
No educational degree	1.77 (1.12–2.78)	0.01	1.41 (0.76–2.62)	0.28
Certificate course	1.44 (0.97–2.15)	0.07	1.43 (0.81–2.52)	0.22
Diploma degree	1.03 (0.69–1.52)	0.90	0.88 (0.51–1.53)	0.66
Bachelor/Master degree	1.00		1.00	
Experience of the trainer ^B^				
No or ≤1 year	3.48 (1.97–6.15)	<0.001	2.53 (1.18–5.44)	0.02
1–5 years	1.33 (1.05–1.69)	0.02	1.12 (0.82–1.54)	0.48
>5 years	1.00		1.00	

^1^ Adjusted for age, gender, BMI, occupation, type of exercise, and training-related factors (i.e., workout knowledge, overweight lifting, and wrong holding). ^A^ Certificate course: having a fitness-related authorized certificate/diploma, 1–3 years degree related to fitness training, bachelor’s/master’s: 4-year bachelor’s degree/1-year master’s degree. ^B^ Working in a fitness center as a trainer.

**Table 3 sports-10-00129-t003:** Risk Factors of the Pain with the Degree and Experience of the Trainer (Generalized Linear Model).

Variables	Unadjusted		Adjusted ^1^	
	IRR (95% CI)	*p*-Value	IRR (95% CI)	*p*-Value
Education of the trainer ^A^				
No educational degree	1.07 (0.82–1.41)	0.581	0.92 (0.68–1.26)	0.637
Certificate course	1.32 (1.01–1.72)	0.039	1.16 (0.85–1.57)	0.332
Diploma degree	1.33 (0.98–1.80)	0.059	1.08 (0.76–1.53)	0.642
Bachelor/master degree	1.00		1.00	
Experience of the trainer ^B^				
No or <1 year	1.95 (1.47–2.57)	<0.001	2.04 (1.50–2.79)	<0.001
1–5 years	1.21 (1.03–1.41)	0.020	1.26 (1.05–1.51)	0.011
>5 years	1.00		1.00	

^1^ Adjusted for age, gender, BMI, occupation, type of exercise, and training-related factors (i.e., workout knowledge, overweight lifting, and wrong holding). ^A^ Certificate course: having a fitness-related authorized certificate/diploma, 1–3 years degree related to fitness training, bachelor’s/master’s: 4-year bachelor’s degree/1-year master’s degree. ^B^ Working in a fitness center as a trainer. IRR = Incidence rate ratio

## Data Availability

The dataset is available upon request.

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
