# Peer review of "Fitness Trainers’ Educational Qualification and Experience and Its Association with Their Trainees’ Musculoskeletal Pain: A Cross-Sectional Study"

_sports, 2022, doi:10.3390/sports10090129_

Round 1

Reviewer 1 Report

Dear authors,it is an interesting article but it’s important to add to the weaknesses of your study that you only selected “the lead trainer of the fitness center whose work experience was a minimum of one month.” It’s also important to add something about that in the discussion, because it also means that many of the trainers in the centres had less experience which might have a negative effect on the incidence of musculoskeletal pain and injuries

Author Response

Please find the authors' answers to the comments in the attachment.

Reviewer 2 Report

I am reviewing the article “Fitness trainers’ educational qualification and experience and its association with their trainees’ musculoskeletal pain: A cross-sectional study.”. The manuscript under consideration is an interesting article on an important topic. However, there are a few minor concerns.

1. What is Musculoskeletal Pain? Additional explanation is needed with references provided.

2. A detailed explanation is needed as to why there is a need to investigate the association between trainer experience and musculoskeletal pain.

3. Regarding the participants in this study, to what extent are they affected by the COVID-19 pandemic? Do the participants have any history of COVID-19?

4.  Since there are many abstract considerations, I would like to see more evidence-based considerations.

Author Response

(The authors gave the same response as above.)

Reviewer 3 Report

Thank you for the opportunity to review your manuscript,  Fitness trainers’ educational qualification and experience and its association with their trainees’ musculoskeletal pain: A cross-sectional study.

I think the inclusion and exclusion criteria should be worded more clearly.

The section on sample size calculation is not clear. It is unknown whether the calculation is for the coaches or the participants.

It is unclear what the data distribution is and where the information is collected. Should better explain this section.

The dimensions of the questions on training and experience are precise, but it is not explained what kind of questions were asked to their trainees.

Should reword the section on independent variables? The variables to be studied and their possible options should be more precise.

If the validity date of the Nordic musculoskeletal questionnaire is already established, please provide details and include a bibliography.

The title "Potential confounding factors" is misleading when it is found in table 1, named "Trainees’ demographic characteristics". This title is misleading.

In the ethical statement section, there is no mention of who signed the consent form for minors.

I believe that including a flow chart would improve the understanding of the findings, the centres of care and the number of each centre.

It does not explain why the selection of <1, 1 to 5 years or more than five years. Is it a random segmentation?

Line 259-260. This statement is not referenced and is not part of the objective of the study. It is not appropriate for inclusion in the discussion.

Author Response

(The authors gave the same response as above.)

Round 2

Reviewer 3 Report

All my questions have been answered. The manuscript has improved considerably. I congratulate the authors for their efforts in the fieldwork and for improving the manuscript.